# Antimony Mining from PET Bottles and E-Waste Plastic Fractions

**Ayah Alassali** [1,*] , **Caterina Picuno** [1] , **Hanin Samara** [2] , **Sascha Diedler** [1] , **Silvia Fiore** [3] and **Kerstin Kuchta** [1]

1   Institute of Environmental Technology and Energy Economics, Waste Resources Management, TUHH—Hamburg University of Technology, Harburger Schloßstr, 36, 21079 Hamburg, Germany
2   Department of Mechanical Engineering, University of Jordan, Queen Rania St., Amman 11942, Jordan
3   DIATI (Department of Environment, Land and Infrastructure Engineering), Politecnico di Torino, corso Duca degli Abruzzi 24, 10129 Turin, Italy
*   Correspondence: ayah.alassali@tuhh.de

**Abstract:** In this study antimony concentration was analyzed in 30 plastic items (from polyethylene terephthalate (PET) bottles and e-waste) directly by X-ray fluorescence spectroscopy (XRF) spectroscopy. PET samples were digested in a microwave oven with aqua regia. The plastic components deriving from e-waste followed three parallel routes: 1. microwave digestion using different acids (aqua regia, 18 M $H_2SO_4$, 12 M HCl and 6 M HCl); 2. conversion into ash (at 600 °C) and then microwave digestion with aqua regia, and 3. extraction with 12 M HCl at room temperature for different durations (2 h and 24 h). Results showed that antimony extraction yields from PET were between 57% and 92%. Antimony extraction from e-waste plastics was more challenging: aqua regia was inefficient for poly (acrylonitrile butadiene styrene) (ABS) samples (extraction yield was about 20% only), while on a mixture of ABS and polycarbonate (PC), aqua regia, $H_2SO_4$ and HCl exhibited equivalent performances (~21%). Ashed samples returned yields ranging from 20% to over 50%. Room temperature extraction on e-waste plastics obtained lower extraction efficiencies, yet longer incubation durations lead to higher yields. In conclusion, the main challenge associated with antimony mining from plastic waste could be its heterogeneous composition; therefore, the development of reference analytical procedures is highly needed.

**Keywords:** antimony; e-waste; extraction; PET; plastic; recovery

## 1. Introduction

In recent years, antimony (Sb) has increasingly become a critically exhaustible resource. Although it was originally classified as a scarce resource with a low concentration in the earth's crust, the amount of Sb in the environment has been increasing due to excessive mining activities, burning of fossil fuels and industrial production of commodities such as textiles, plastics, electronics, ceramics, glass decolorizers, lead batteries and flame retardants [1]. Antimony was identified by the European Commission (EC) as a critical raw material (CRM), on which Europe's economy is highly dependent [2]. China is currently dominating the global production of Sb; in 2015 it accounted for about 77% of the world's total production [3]. This in turn results in EU's dependence on China for the supply of Sb, and consequently stirs concerns regarding the disruption of its secure and sustainable supply. It is expected that Sb will reach its highest "supply and demand deficit" over the period 2015–2020 [2]. Extractable global reserve of Sb was estimated in 2008 by the United States Geological Survey (USGS) to be around 2.1 million tons [4], while in 2016 it decreased to about 1.5 million tons [5]; this shows about a 30% reduction in the global recoverable reserve in less than a decade and in turn asserts the rapid exhaustibility of Sb.

Consequently, it has become crucial to improve the consumption efficiency of such a resource and to effectively identify the conditions for its recyclability. Accordingly, and in line with its circular economy action plan (COM/2015/0614), the EC has mandated recycling to reduce the risk of insufficient supply of Sb and other CRMs [2]. The end-of-life recycling input rate (EOL-RIR)—defined as the ratio of the amount of CRM recovered from recycling to the total amount of material's input into production—of Sb in the EU is estimated to be around 28% (see Figure 1) [6]. The low EOL-RIR can be attributed to many factors, such as the absence of sorting and recycling technologies at competitive costs and the retention of Sb in many long-life assets such as lead alloys that are used in construction. Additionally, the EOL-RIR is highly affected by in-use dissipation, which results in a reduction in materials availability for recovery and reuse, greater dependence on primary geological resources, in addition to increased settling of Sb in various ecosystems [7,8]. Heckens et al. (2016) proposed an extraction rate reduction of 94% from those of the year 2010 through efficient Sb use, increased recycling and finding suitable substitutes of such resource in applications where it is used [9].

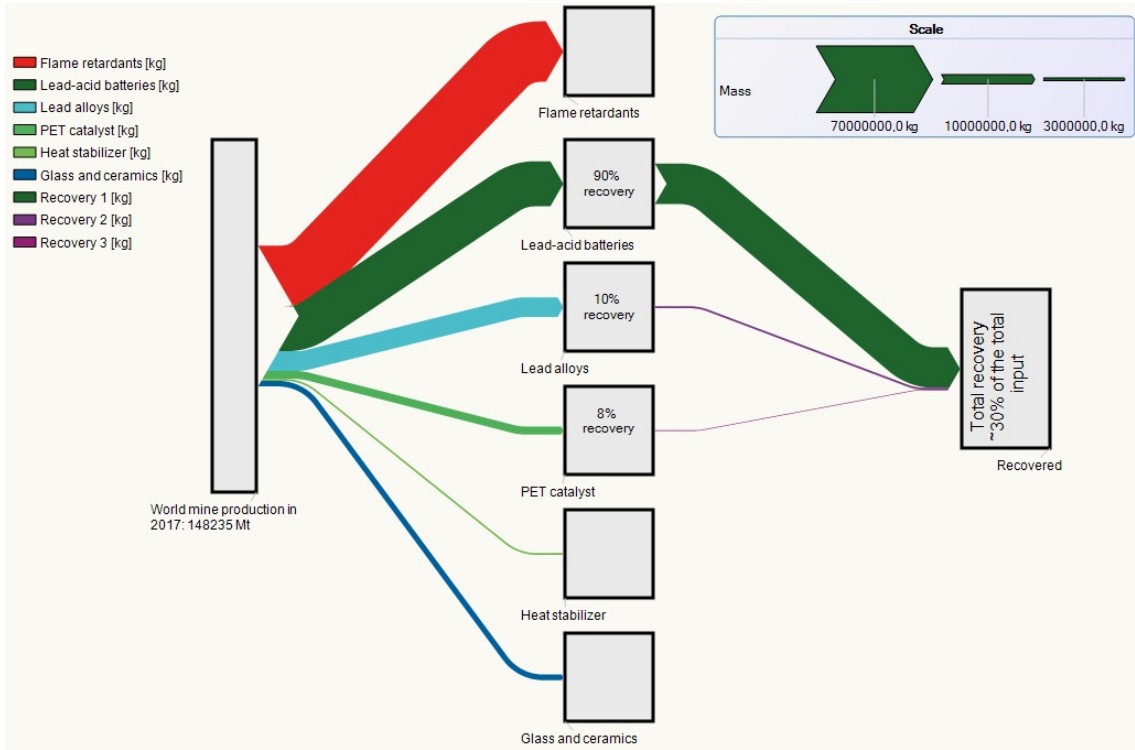

**Figure 1.** An overview of antimony (Sb) utilization in different industrial applications [9–11].

The global consumption of Sb in 2018 was estimated as follows [11] (Figure 1): 43% in flame retardants, 46% in lead batteries and lead alloys, 6% in catalysts for the production of polyethylene terephthalate (PET) bottles, and 5% for chemicals, glass and ceramics. However, and as a consequence of the dissipative distribution of Sb, no recycling processes exist on a technical scale for most fields of application [12], except for lead-acid batteries [9]. As a consequence, the global functional recycling rate of non-metallurgical applications at present ranges from 1% to 10% [13,14], indicating an ineffective recovery. It was reported that secondary Sb sources (e.g., fly ashes, slags from lead refineries and scraps), industrial by-products (e.g., dump material, waste from metals extraction) and other activities (e.g., landfills or residues from waste incineration) are of increasing interest for its recovery [12].

Antimony is present in many plastic materials, mostly as synergetic flame-retardant, and for this reason it is found within the waste cycle [15]. A recent study revealed the ubiquity of Sb in plastics used in various applications [16]. In another study [17], different types of acids were tested to assess their efficiency in leaching Sb from a discarded computer housing, made of poly (acrylonitrile butadiene

styrene) (ABS). Results showed that the leaching medium (i.e., heated solution of sodium hydrogen tartrate in dimethyl sulfoxide) could leach up to 50% of the Sb from ABS.

On one hand, numerous efforts have been done to assess the concentration of Sb in plastics (e.g., [16]), on the other, very few studies regarding Sb extraction from different polymers for the aim of recycling are available [17,18]. Moreover, reference analytical procedures are completely missing. This work was focused on plastic components of e-waste, containing relevant amounts of Sb [19], and on PET bottles, chosen because of their high abundance in plastic waste flows [20] and significant Sb content (up to 200–350 mg/kg) [21–23]. E-waste management should be heavily improved to achieve EU targets [24] and to limit its environmental impacts [25]. Furthermore, in 2007, 24% of PET were globally recycled; 72% of recycled PET flakes were utilized in polyester fiber and 10% reused in bottles [26]. Meaning that, from both plastic waste flows, the recycled material will potentially mobilize Sb into different production cycles as well as into the environment. Therefore, easy and economical methodologies of extraction need to be investigated for Sb mining from plastic waste.

With these premises, this research aims at giving an assessment of Sb recovery through acid extraction from e-waste plastic fraction and PET bottles. This would, on the one hand, fill the knowledge gaps in the current literature on Sb extraction from different polymers and, on the other hand, provide a first basis for further developing the extraction methodology eventually leading to the development of large-scale application. Firstly, the different polymers were identified, and the Sb content was directly measured in all samples (30 items) through X-ray fluorescence spectroscopy (XRF). Afterwards, different types (aqua regia, hydrochloric and sulfuric acids) and concentrations of acids have been investigated as leaching agents, in addition to microwave-assisted extraction. Plastics derived from e-waste underwent microwave-assisted acidic extraction, both unaltered and after their conversion to ashes at 600° C. Plastics from e-waste also underwent room-temperature acidic extraction, to evaluate the extraction yield towards Sb, as well as the effects on the plastic items. This work had two objectives: 1. to investigate Sb recovery from waste plastics and their ashes, thus hypothesizing different waste management routes (separate collection of PET and of plastics from e-waste); 2. to evaluate the possibility to convert plastic waste into a non-hazardous fraction, which can be easily returned into the production cycle after Sb extraction, or disposed without further treatment.

## 2. Materials and Methods

### 2.1. Samples Origin

Two materials were considered in this study: PET from bottles and mixed plastics from e-waste, due to the presence of Sb in these two waste streams [19,21]. Virgin PET granules from NEOPET 82 FR—INEOS Olefins and Polymers Europe were considered as reference material. PET samples (about 300 g each) had three different origins:

-    washed and shredded PET flakes, prepared for recycling (provided from a bottle-to-bottle recycling facility in Germany),
-    PET from recyclable soft-drink bottles,
-    PET from water bottles.

As for the plastic fraction from e-waste, the samples were obtained from an e-waste collection point in Hamburg (Germany). The analyzed items were 27 in total: external hard plastic casings of 20 mobile phones, external casings of five laptops, an external casing of one PC screen, and an external casing of one TV screen.

### 2.2. Samples Preparation

PET samples from soft-drinks and water bottles were cut with scissors into pieces (size smaller than 5.0 mm × 5.0 mm). E-waste plastic parts were manually dismantled and qualitatively analyzed by near infrared (NIR) spectroscopy using a Thermo Fisher Scientific™ microPHAZIR™ RX Analyzer.

Each of the identified polymers was singly shredded to a particle size ≤ 4.0 mm in two steps, first using a JBF universal shredding machine, secondly, through a SM 300 Retsch Cutting Mill.

### 2.3. Total Antimony Concentration Assessment

The total Sb concentration in the considered samples was measured using XRF (Thermo Fisher Scientific™ Niton™ XL3t XRF-Analyzer, MA, USA. The normal filter was applied to analyze Sb in the 'plastics mode', based on characteristic peak at 26.359 KeV (Kα). The XRF was used in a collapsible bench-top accessory stand (Thermo Scientific SmartStand), with a stainless-steel window-surrounding base and directly connected to a computer via USB. Analyses were done in triplicates. A Multi-element Reference Sample EN 71-3, LOT # T-60 was analyzed as a reference sample for cross-validation.

### 2.4. PET Samples Acidic Extraction

An acidic extraction according to DIN EN 16174 procedure was performed (0.5–1.0 g sample was digested in 10 mL of aqua regia) by a CEM Mars 6 microwave oven (175 °C for 15 min at 800 psi). Acidic extracts were filtered on 1.2 μm membranes and diluted to 100 mL with deionized water.

### 2.5. E-Waste Plastics

In order to assess the Sb content in e-waste plastics, the samples were analyzed in their original form and as ashes (see Figure 2), hypothesizing that e-waste plastics could be both separately collected for recycling or incinerated.

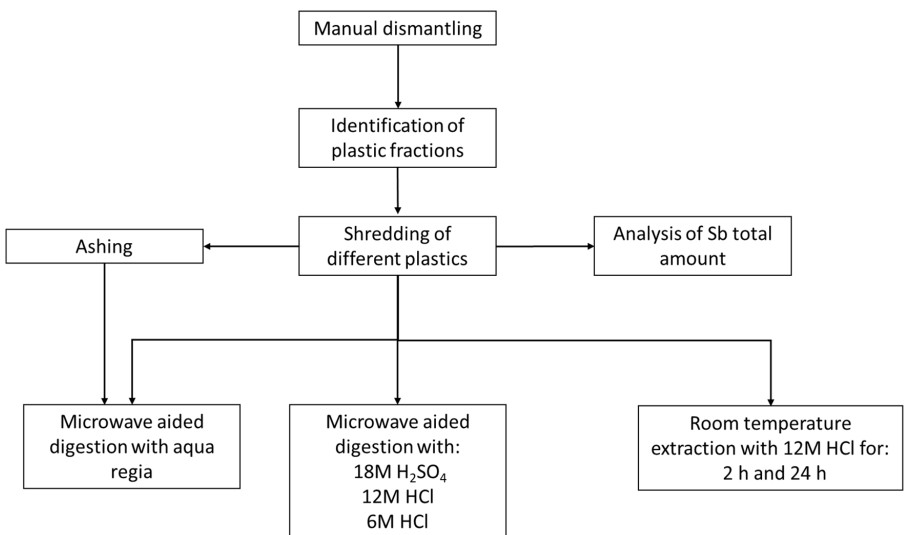

**Figure 2.** Experimental procedure implemented on e-waste plastic samples.

### 2.5.1. Ashing

The ISO standard 3451-1 procedure A [27] was followed to evaluate the ash content in samples obtained from e-waste plastic specimens. The samples were weighed and placed inside porcelain crucibles, then pre-burned with a butane Bunsen burner. Afterwards the samples were converted into ashes in a muffle at 600 °C. The samples were taken out from the muffle every half hour, cooled in a desiccator and weighed. This step was repeated until a constant mass was obtained (total time not exceeding 3 h) [28].

### 2.5.2. Acidic Extraction

Preliminary tests performed on e-waste plastic components proved aqua regia as an ineffective acidic extraction medium [28]. Therefore, with the aim of investigating other leaching phases, e-waste plastic samples were microwave-digested using different acids: 6 M HCl, 12 M HCl and 18 M $H_2SO_4$,

adopting the conditions detailed in Section 2.4. Ashed samples (0.5–1.0 g) were also microwave-digested in duplicates adopting the same conditions detailed in Section 2.4.

To compare the results obtained by applying microwave extraction to less expensive and less harsh methods, a milder extraction was tested solely on e-waste plastic samples. Triplicates of 0.5–1.0 g samples were soaked in 10 mL of 12 M HCl at 20 °C. In order to assess the degree of influence of the digestion time, two incubation durations were chosen; 2 h and 24 h. At this point, it is worth mentioning that $Sb_2O_3$ is not very soluble, and non-oxidizing acids are not able to react with Sb unless $O_2$ is present [29]. The use of HCl was hypothesized since HCl is used extensively in industrial applications [30] and analysis of environmental samples [31,32].

### 2.6. Antimony Analysis After Acidic Extraction

The acidic solutions obtained from the different extraction trials were vacuum-filtered through glass fiber with a pore size of 1.2 μm, and then diluted to 100 mL with distilled water. The analysis of Sb concentration was performed through an Agilent 5100 inductively coupled plasma atomic emission spectroscopy (ICP-OES), after a calibration with ICP-OES multi-element calibration standard-3 from Agilent (part number 8500-6948).

## 3. Results

### 3.1. Antimony Recovery from PET

The direct assessment of Sb concentration in PET samples revealed a total content ranging between 231.3 and 256.8 mg/kg. As shown in Figure 3, 71.4 ± 6.3% of the total Sb could be extracted from virgin PET, PET from the water bottles achieved the highest yield (92.5 ± 16.4%), and PET from soft drinks bottles showed a wide variation in the extraction yield (78.3 ± 47.4%). PET flakes originating from the recycling facility exhibited the lowest yield with a significant variation (56.6 ± 33.6%).

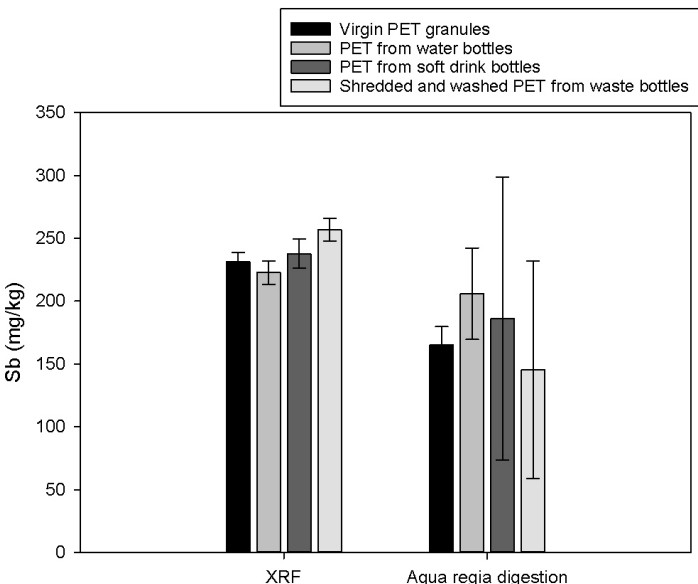

**Figure 3.** Antimony concentration in PET samples: direct measurement through XRF and microwave assisted digestion with aqua regia and ICP-EOS analysis of the extract.

### 3.2. Antimony Recovery from E-Waste Plastics

#### 3.2.1. Antimony Total Amount in Shredded Samples

The XRF analysis (Figure 4) indicated minor concentrations of Sb in the mobile phone external covers (25.4 ± 15.3 mg/kg), with a relevant variability among items of different models and brands.

The same trend was observed in the PC screen (Sb concentration 21.7 ± 11.7 mg/kg). On the other hand, Sb was absent in the TV screen plastic fraction (concentrations below the detection limits 14 ppm). An elevated Sb concentration was detected in laptop casings (152.5 ± 64.9 mg/kg).

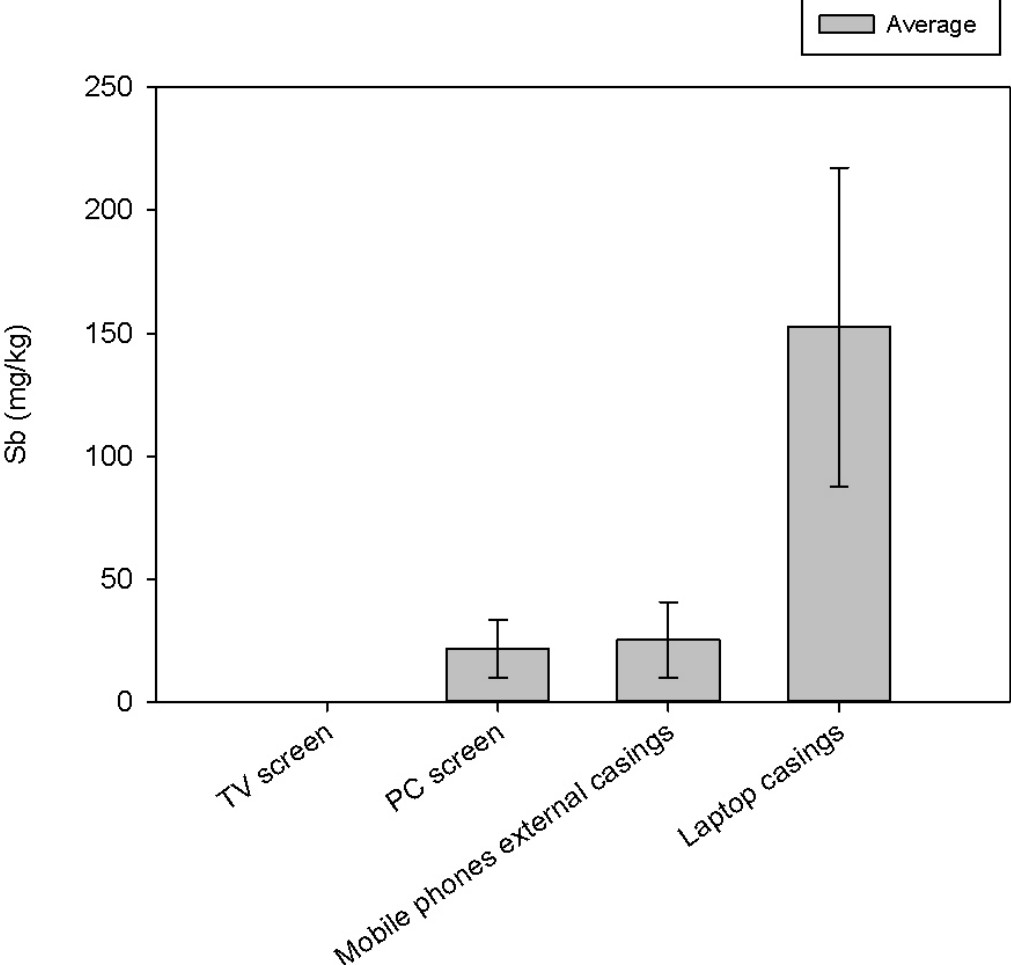

**Figure 4.** Antimony concentrations in the considered samples (direct measurement by XRF).

### 3.2.2. Antimony Extraction from Ashes

As regulated in the ISO 3451-1 standard [27], the process of ashing was conducted until a constant sample weight was reached, with a maximum duration of 3 h. The sample mass steadiness was reached earlier by the samples derived from the TV screen (Figure 5b) and the PC screen (Figure 5d). The ash content in laptop casings, mobile phone casings, the TV screen and the PC screen was 4.48 ± 0.10%-wt, 4.62 ± 1.66%-wt, 0.47 ± 0.02%-wt and 1.37 ± 0.02%-wt, respectively.

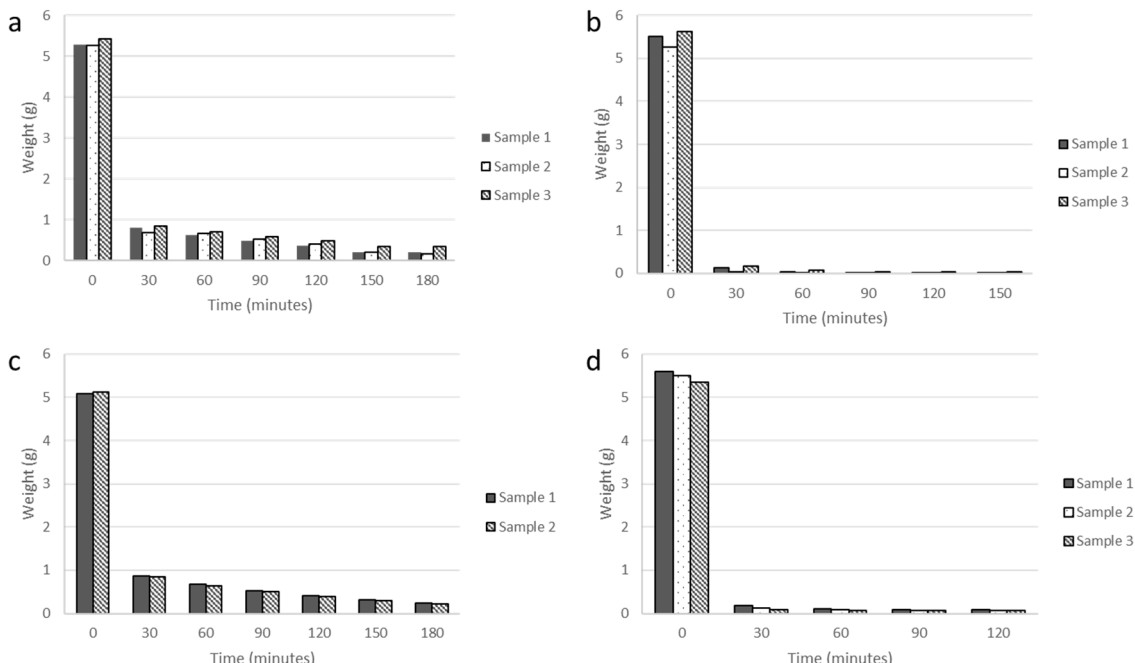

**Figure 5.** Sample weight change during the ashing process: (**a**) mobile phones casings, (**b**) TV screen, (**c**) laptop casings and (**d**) PC screen.

### 3.2.3. Comparison of Extraction Yields

Figure 6 shows a comparison of the extraction yields achieved through the examined extraction procedures. Results showed very low concentrations of Sb extracted from the TV screen ($\leq 8.40 \pm 2.8$ mg/kg), with the highest value achieved using 18 M $H_2SO_4$. Since the total Sb concentration was below the XRF detection limit (14 ppm), extraction yields could not be calculated for the TV screen.

The Sb extraction yields from the PC screen were as follows: 0.7% with aqua regia, 19.4% with 18 M $H_2SO_4$ and 21.5% with 12 M HCl. On the other hand, 50.1% of Sb recovery was obtained from the ash form. Extraction at room temperature with 12 M HCl showed that longer durations obtained higher extraction yields; yields were 10 folds of what was obtained by aqua regia when extraction lasted 2 h and 20 times higher after 24 h of incubation.

For mobile phone casings, comparable extraction yields were obtained by aqua regia ($12.6 \pm 10.4\%$), 18 M $H_2SO_4$ ($13.3 \pm 15.6\%$), and 6 M HCl, ($15.9 \pm 10.5\%$). Digestion using 12M HCl obtained the highest yield ($21.2 \pm 13.5\%$), where material ashing indicated high losses in the fly ash fraction (about 80%). Raising the extraction duration from 2 h to 24 h at room temperature resulted in an extraction yield increase by 42.7% (from 5.2 to 7.2%), yet the yields were insignificant.

The extraction of Sb from laptop casings showed elevated Sb yields, exceeding 100%. Nevertheless, applying the standard deviation values in the assessment provided within-the-limits yields, except for the extraction done using 18 M $H_2SO_4$. Conversely, extraction at room temperature was unable to recover Sb from the laptop casings.

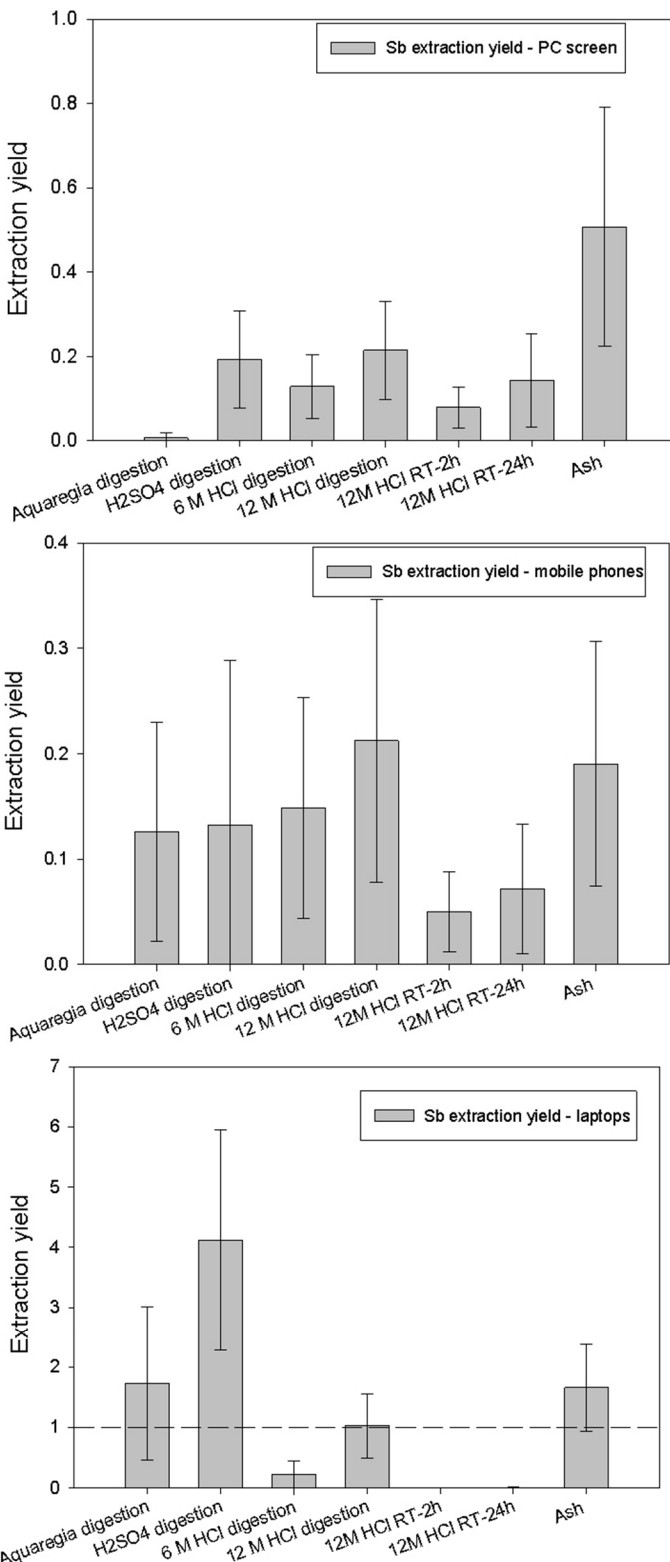

**Figure 6.** Antimony extraction yields from the considered plastic waste samples.

## 4. Discussion

This paper analyzes possible methods to address Sb recovery from plastic waste streams, for the aim of formulating practical approaches for Sb recycling. Microwave-assisted digestion has often been used to decompose polymers [22,23,33]. Hence, this method was considered for Sb extraction on all the shredded plastic samples. PET samples were only digested with aqua regia, because the extraction procedure immediately proved highly effective, as explained in the following sections.

### 4.1. Antimony Recovery from PET

The assessment of Sb concentration in PET samples revealed a total content that is in agreement with literature, indicating the effectiveness of XRF in providing the total concentration of Sb in the samples. Jesus et al., (2016) reported Sb concentration values in PET bottles in the range between 194 and 323 mg/kg, analyzed by means of graphite furnace atomic absorption spectrometry (GF-AAS) [21], whereas Chapa-Martínez et al. (2016) used hydride generation atomic fluorescence spectrometry (HG-AFS) to detect a Sb concentration ranging between 73 and 111 mg/kg [34]. Takahashi et al., (2008) analyzed 177 PET bottles from China and Japan using X-ray absorption fine structure (XAFS), and Sb concentrations ranged between 10 and 350 mg/kg [22]. Westeroff et al., (2008) reported similar concentrations (213 ± 35 mg/kg) from PET bottles [23].

The wide variation in the extraction yield of Sb from soft drinks bottles could be attributed to the variation in the original concentration in the bottles (they are claimed to be partially containing recycled material). The low Sb extraction yield from the PET flakes originating from the recycling facility could be explained by the heterogeneous concentration of Sb in the flakes as a result of handling various types of products and/or by losses during the washing and drying steps of the recycling process. Moreover, the nature of the shredded and washed PET flakes is more brittle in comparison to the flakes of the water bottle purchased from the market, which could have hindered efficient and consistent Sb extraction. In general, it may be noticed that acidic extraction is a mass transfer process happening through the contact surface between the plastic waste and the extracting solvent; therefore, a higher variation in the experimental results if compared to the direct measurement performed with the XRF spectrometer was expected.

### 4.2. Antimony Recovery from E-Waste Plastics

#### 4.2.1. Antimony Total Amount in Shredded Samples

As indicated in Figure 4, Sb was absent in the plastic fraction of the TV screen, indicating the possibility of using different types of flame-retardants [35].

Generally, e-waste plastic samples exhibited lower Sb concentrations if compared to PET samples (Figure 3), except for the laptop casings. This could be attributed to the production date of these items, where Sb might have not been extensively used. This was confirmed by the high, yet inhomogeneous flame retardants tracer concentrations [36] (i.e., Br + Cl and P), showing that other types of flame retardants were present (Table 1).

**Table 1.** Br + Cl and Phosphorous (P) concentration in the analyzed samples.

| Sample | Br + Cl (ppm) * | P (ppm) ** |
|---|---|---|
| TV screen | 417.0 ± 292.8 | 2.4 ± 1.7 |
| PC screen | 4.9 ± 0.3 | 103.1 ± 2.9 |
| Mobile phones | 134.8 ± 0.7 | 410.9 ± 8.2 |
| Laptops | 60.1 ± 56.0 | 2755.2 ± 299.4 |

* Analysis done by XRF. ** Analysis done on the ash fraction using ICP-EOS after extraction with aqua regia.

### 4.2.2. Antimony Extraction from Ashes

Before conducting sample ashing, a pre-ashing step was applied to control the losses which could happen by fast burning (leading to ignition), during which both laptop and mobile phone casings needed longer time under the Bunsen burner to stop generating fumes (see Figure 5). For laptop casings, the fuming was accompanied by liquefaction of the casings, indicating a high concentration of metals [13]. The ash content in laptop casings and mobile phone casings was higher than that in the TV screen and in the PC screen. These results were consistent with the behavior obtained during the pre-ashing step, which predicted higher metallic content in laptop and mobile phone plastic casings. Moreover, and as clearly seen in Figure 5, mass steadiness was reached earlier by the samples derived from the TV screen (Figure 5b) and the PC screen (Figure 5d), indicating their higher organic content and, respectively, their lower metal content in comparison to laptop and mobile phone casings.

### 4.2.3. Comparison of Extraction Yields

The extraction of Sb from laptop casings presented Sb yields exceeding 100%, especially for the extraction done using 18 M $H_2SO_4$. These exceeding yields (beyond 100% total Sb) might have happened due to material contamination with laptop inner metals during shredding.

### 4.3. Evaluation of Different Extraction Solvents and Conditions

Different extraction solvents have been tested in this work with the aim of optimizing the extraction of Sb from e-waste plastic fraction. The samples composition was detected as follows: ABS in TV screen and PC screen, mobile phone casings were mostly made of ABS, polycarbonate (PC) and a mix of both, and laptops were a mixture of PC/ABS and ABS.

As shown in Figures 6 and 7, the extraction yields by different acidic solvents varied between the samples. This could be accredited to the different polymers present in the samples as well as to the various types of additives and coatings. For instance, extraction using aqua regia showed the least extracted yields of Sb for both the TV screen and the PC screen (which were entirely composed of ABS). When the average extraction values and the standard deviations are included in the evaluation, comparable extraction yields were obtained by 18 M $H_2SO_4$, 12 M HCl and 6 M HCl, indicating that aggressive acids did not improve the extraction from ABS polymer. Furthermore, incubating the samples in 12 M HCl at room temperature could not achieve Sb extraction from the TV screen, yet incubating the PC screen at room temperature for 24 h retained concentrations similar to what was obtained by microwave-assisted digestion, proving that HCl is efficient in impairing ABS. For the mobile phone casings, all acids achieved similar extraction yields, while the lowest yields were obtained by 12 M HCl at room temperature. Nevertheless, when such low concentrations of Sb are analyzed in the plastic fraction of e-waste units (the mobile phonecasings in this case), the material would not be considered feasible for Sb extraction. Laptop casings did not follow the same trend, where digestion using 18 M $H_2SO_4$ provided the highest extraction, followed by aqua regia (about the half of what was extracted by 18 M $H_2SO_4$). 12 M HCl showed significantly higher extraction yields in comparison to 6 M HCl (factor of increase was around 4.5), when microwave digestion was applied, yet, the extraction at room temperature could not solubilize any Sb.

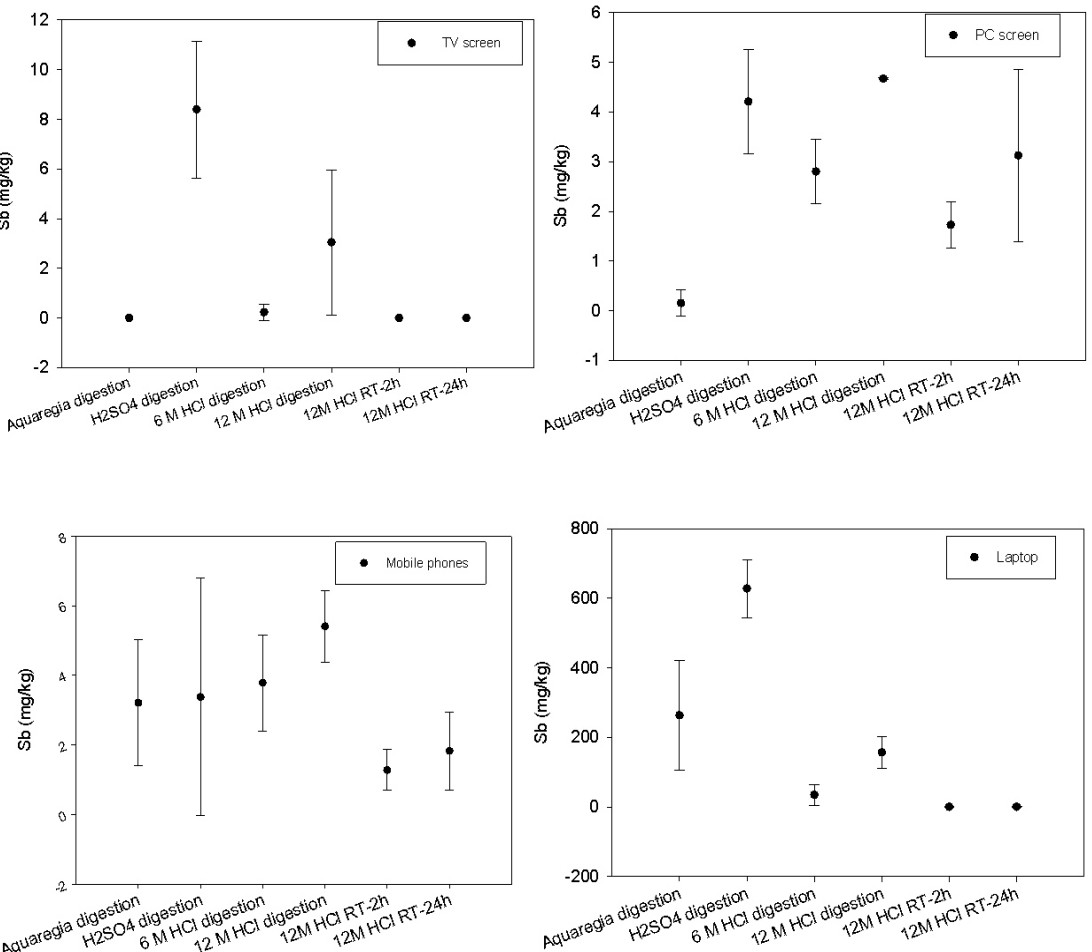

**Figure 7.** Comparison of the efficiencies of different acids and extraction conditions towards Sb extraction from e-waste plastic fractions.

## 5. Conclusions

Due to the current supply risk of raw materials, there is an increasing interest in advancing technologies for critical material recovery and recycling [37,38]. This study focused on optimizing extraction procedures to recover the critical metal Sb; through assessing different extraction solvents, concentrations and parameters. The antimony content screening performed on the considered samples showed that not all plastic samples derived from e-waste contained significant amounts of Sb, suggesting that other types of flame-retardants were probably used. Direct measurement (through XRF spectroscopy) on the unaltered samples and the analyses of the acidic extracts from ashed samples proved this theory; TV screen contained 417.0 ± 292.8 mg/kg Br + Cl, PC screen contained 103.1 ± 292.8 mg/kg Br + Cl, mobile phone external covers contained 410.9 ± 8.2 mg/kg Br + Cl and 134.8 ± 8.2 mg/kg P and laptopexternal casings contained, in addition to the high Sb content, high phosphorous content (2755.2 ± 299.4 mg/kg).

PET exhibited the highest extraction yields (from 56.6% to 92.5%), most likely due to the thin elastic material of which the bottles are composed, which lacks coatings and additives.

For the plastic samples tested from e-waste units, aqua regia was unable to extract Sb from ABS, while 18 M $H_2SO_4$ and 12 M HCl were more efficient. For the items composed of ABS/PC and PC, Sb extraction yields were similar with aqua regia, 18 M $H_2SO_4$ and 12 M HCl. 6 M HCl returned lower yields from the laptop casings, yet the yields were intermediate in the mobile phone casings. For extraction at room temperature, only the PC screen and the mobile phone casings returned Sb, and longer durations were more efficient. For future research activities, a wider range of plastic samples

from various units, polymer types and production dates should be tested, to propose a universal extraction method for Sb mining from plastic waste streams.

**Author Contributions:** Conceptualization, A.A., S.F., and K.K.; Methodology, A.A.; Validation, A.A., C.P. and S.F.; Formal Analysis, A.A.; Investigation, A.A and C.P.; Resources, H.S.; Data Curation, A.A., and C.P; Writing—Original Draft Preparation, A.A., C.P., H.S., and S.D.; Writing—Review and Editing, S.F., S.D., and K.K.; Supervision, S.F. and K.K.

**Funding:** This research did not receive any specific grant from funding agencies in the public, commercial, or not-for-profit sectors.

**Acknowledgments:** The authors would like to thank Piergiorgio Gazzellone for supporting the experimental activities.

**Conflicts of Interest:** The authors declare no conflict of interest.

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
