# Peer review of "Antimony Mining from PET Bottles and E-Waste Plastic Fractions"

_sustainability, doi:10.3390/su11154021_

Round 1
Reviewer 1 Report
General comments
I want to congratulate the authors for their extensive experimental work.
The study was aimed to give an assessment of antimony recovery through acid extraction from e-waste plastic fraction and PET bottles. Some of different types (aqua regia, hydrochloric and sulfuric acids) and concentrations of acids have been investigated as leaching agents.
The paper is complex and responds to the topics of the journal. The experiments refer to materials that are very used in the recent years. The structure and the composition of the paper is good, but I found some issues that must be solved by the authors.
Many standards have been chosen for the experiments and the appropriate form was used in all the cases.
Observations/questions
· Figure 5 presents a comparison for the ash content determined for 4 types of e-waste. I recommend to authors to release a comparative idea based on the obtained results.
· 26 of references were used in Introduction of the paper, another 6 are mentioned in Materials and Methods and only 2 references are used for the Results and Discussions. This is a serious minus for the paper. So, I recommend to the authors to improve this aspect and give more examples from the scientific literature that strengthens the experimental results presented in the present paper.
Author Response
Dear Editor, dear Reviewer, Thank you for your comments and feedback. Below are the answers for the raised issues. The line numbers cited in the following refer to the “revised with changes” version of the manuscript. The changes are highlighted in the “revised with changes” version of the manuscript.
Comment 1: Figure 5 presents a comparison for the ash content determined for 4 types of e-waste. I recommend the authors to release a comparative idea based on the obtained results.
Response: The comparative idea based on the results is provided in the discussion; please see section 4.2.2 (Lines 289 – 292).
Comment 2: 26 of references were used in Introduction of the paper, another 6 are mentioned in Materials and Methods and only 2 references are used for the Results and Discussions. This is a serious minus for the paper. So, I recommend to the authors to improve this aspect and give more examples from the scientific literature that strengthens the experimental results presented in the present paper.
Response: This is correct. Yet some of the references, which were used in the introduction and materials and methods sections, were re-utilized in the discussion section, due to their relevance to the outcomes of the study. Four more references were added: see No. 8, 36, 38 and 39 in Reference list (in the text see lines 56, 271, 329). The total references used in the results and discussion and conclusion sections were 13.
Reviewer 2 Report
The proposed research «Antimony mining from PET bottles and e-waste plastic fractions » falls within the scope of Sustainability. According to the reviewer’s opinion, some revisions are required. Please, comply with the following suggestions and comments:
Comment 1: The paper is in general well accompanied of clear explanations.However, questions that need to be answered: Why your study is important? How it extend the existing knowledge on the issue/topic? Concluding remarks – authors must elaborate more on what is their contribution to the literature.
Comment 2: More recent papers in the field should be integrated in the literature review.
Comment 3: Finally, when you submit the corrected version, please do check thoroughly, in order to avoid grammar, syntax or structure/presentation flaws - please seek for professional English proofreading services or ask a native English-speaking colleague of yours in order to refine and improve the English in your paper.
Author Response
Dear Editor, dear Reviewer, Thank you for your comments and feedback. Below are the answers for the raised issues. The line numbers cited in the following refer to the “revised with changes” version of the manuscript. The changes are highlighted in the “revised with changes” version of the manuscript.
Comment 1: The paper is in general well accompanied of clear explanations. However, questions that need to be answered: Why your study is important? How it extend the existing knowledge on the issue/topic? Concluding remarks – authors must elaborate more on what is their contribution to the literature.
Response: A further elaboration was provided to discuss the aims and novelty of the study, please see Lines 93-96. Also, the objectives are stated in Lines 103 to 108. The reason behind choosing the samples was added (Line 112). Additional remarks are added in Lines 240-241 and 328-331.
Comment 2: More recent papers in the field should be integrated in the literature review.
Response: Recent papers were integrated in both the literature review section as well as the discussion and conclusion sections (please see No. 8, 36, 38 and 39 in Reference list).
Comment 3: Finally, when you submit the corrected version, please do check thoroughly, in order to avoid grammar, syntax or structure/presentation flaws - please seek for professional English proofreading services or ask a native English-speaking colleague of yours in order to refine and improve the English in your paper.
Response: The language was refined and improved throughout the text.